# TiO$_2$ NPs-immobilized silica granules: New insight for nano catalyst fixation for hydrogen generation and sustained wastewater treatment

**Nasser A. M. Barakat** [1]*, **Osama M. Irfan** [2]*, **Olfat A. Mohamed**[3]

**1** Chemical Engineering Department, Faculty of Engineering, Minia University, El-Minia, Egypt,
**2** Department of Mechanical Engineering, College of Engineering, Qassim University, Buraydah, Saudi Arabia, **3** Chemical Engineering Department, Faculty of Engineering, Port Said University, Port Said, Egypt

* nasbarakat@mu.edu.eg (NAMB); o.ahmed@qu.edu.sa (OMI)

## Abstract

In heterogeneous catalytic processes, immobilization of the functional material over a proper support is a vital solution for reusing and/or avoiding a secondary pollution problem. The study introduces a novel approach for immobilizing R25 NPs on the surface of silica granules using hydrothermal treatment followed by calcination process. Due to the privileged characteristics of the subcritical water, during the hydrothermal treatment process, the utilized R25 NPs were partially dissolved and precipitated on the surface of the silica granules. Calcination at high temperature (700˚C) resulted in improving the attachment forces. The structure of the newly proposed composite was approved by 2D and 3D optical microscope images, XRD and EDX analyses. The functionalized silica granules were used in the form of a packed bed for continuous removal of methylene blue dye. The results indicated that the TiO$_2$:sand ratio has a considerable effect on the shape of the dye removal breakthrough curve as the exhaustion point, corresponding to ~ 95% removal, was 12.3, 17.4 and 21.3 min for 1:20, 1:10 and 1:5 metal oxides ratio, respectively. Furthermore, the modified silica granules could be exploited as a photocatalyst for hydrogen generation from sewage wastewaters under direct sunlight with a good rate; 75×10$^{-3}$ mmol/s. Interestingly, after the ease separation of the used granules, the performance was not affected. Based on the obtained results, the 170˚C is the optimum hydrothermal treatment temperature. Overall, the study opens a new avenue for immobilization of functional semiconductors on the surface of sand granules.

**Data Availability Statement:** All relevant data are within the manuscript.

**Funding:** This research has been financially supported by Deanship of Scientific Research,

## 1. Introduction

Photo catalysis is the hottest field having large applications of nanomaterials as photo catalysts especially in water treatment and hydrogen generation from water splitting [1]. In green hydrogen production, photo catalysis can be used to split water molecules into hydrogen and

Qasim University. The funders had no role in study design, data collection and analysis, decision to publish, or preparation of the manuscript.

**Competing interests:** The authors have declared that no competing interests exist.

oxygen, providing a clean and renewable source of energy. The photons from the light excite the electrons in a semiconductor photocatalyst, allowing them and the corresponding formed holes to participate in the reaction. As a result, water molecules are broken down into hydrogen and oxygen. This process is considered green because it does not generate greenhouse gases or other pollutants [2, 3].

In the degradation of organic pollutants, photo catalysis is used to break down organic pollutants in the environment [4–6]. This is particularly useful for pollutants that are resistant to conventional treatment methods, such as persistent organic pollutants (POPs) [7]. Among the reported wastewater treatment processes, photo degradation of the organic pollutants is the simplest and cheapest treatment strategy [8]. In this process, a highly reactive species (e.g. OH radical and $H_2O_2$) are formed due to photons absorption by the semiconductor photo catalysts [9]. Consequently, the photodegradable organic molecules are oxidized and broken down to simple and environmentally safe molecules. This process can be performed both in water and air, making it a versatile tool for environmental remediation [10, 11]. Moreover, photo degradation can also be brought on by other types of electromagnetic radiation [12]. Overall, photo catalysis is a promising technology for both producing green hydrogen and mitigating environmental pollution.

The surface area of the utilized photo catalyst is a key factor. Accordingly, due to the extremely high surface area, the nanomaterials showed excellent performance. The high surface area allows for a larger number of active sites, which increases the rate of reaction. The increased active sites also increase the probability of collisions between the reactants and the catalyst, leading to improved reaction efficiency. In addition to the high surface area, the nanoscale size of photo catalysts also allows for improved light absorption and increased light penetration, leading to more efficient utilization of light energy. Furthermore, the small size of nanomaterials also enhances the mass transport of reactants to and from the surface, leading to a faster reaction rate.

Accordingly, in the field of green hydrogen production from water splitting, several functional nanomaterials offered a distinct performance [13, 14]. In the same fashion, nanomaterials exhibit an excellent performance as photo catalysts in organic pollutants removal by photo catalysis technique [15, 16]. Paradoxically, exploiting the nanomaterials photo catalysts still applicable in a lab scale because most of the researchers ignored separation of the used function nanomaterials from the reactions media. Therefore, in wastewater treatment, indeed, the nanomaterials showed an excellent performance as photo catalysts in dye degradation, these nanocatalysts create a secondary pollution.

Immobilization of these catalysts can address this issue if an appropriate support and a suitable fixation technique are employed [17–19]. This would ensure that the catalyst remains stable throughout its intended use and provide effective results with minimal wastage or loss of material over time [20]. In addition, catalyst immobilization minimizes costs by enabling reuse, increasing stability, preventing catalyst self-destruction, and controlling the reaction media [21]. In this regard, different supports for functional photo catalysts have been investigated including hydrogels, carbon nanotubes [22], polymers [23], silk fibroin [24] and other inorganic materials [25].

Silica is a highly sought-after support material due to its impressive thermal and chemical stability, as well as its large surface area. This makes it an ideal support matrix for numerous functional materials, allowing reactants to access the active sites of the catalyst more easily than with other materials. In this regard, this promised support has been utilized for immobilizing several catalysts such as metalloporphyrins [26], rhodium [27] and methylaluminoxane [28].

In recent years due to their high efficiency and environmental sustainability, $TiO_2$ NPs drew the most attention as an effective photocatalyst [29, 30]. These semiconductor nanoparticles have been widely used as photo catalysts due to several advantages including high photocatalytic activity, non-toxicity and high stability toward heat, chemical reactions, and environmental degradation which making $TiO_2$ suitable for use in a wide range of applications. Moreover, $TiO_2$ is a widely available and low-cost material, making it a suitable choice for large-scale environmental cleaning applications [31]. In addition, this fantastic semiconductor can be modified with various materials to improve its photocatalytic performance and make it suitable for different applications [32–34]. As a result, $TiO_2$–based nanoparticles are the most widely used photo catalysts in wastewater treatment and green hydrogen production that creates an important step towards a more sustainable and environmentally friendly future. However, to our best knowledge, immobilization of this highly important semiconductor has not yet been introduced.

In this study, R25 (commercial grade of $TiO_2$ NPs) was successfully immobilized on the surface of silica granules using the high solubility power of the subcritical water[35, 36]. Typically, hydrothermal treatment of R25 and silica granules resulted in partial solubility of the semiconductor NPs and deposition on the surface of the proposed support. Calcination of the coated granules was performed to enhance the adhesion forces between the two oxides. The activated silica granules showed good activity as a packed bed for continuous dye removal as well as in wastewater photo splitting for hydrogen production.

## 2. Experimental part

### 2.1 Materials and methods

Specific amounts from $TiO_2$ nanoparticles (R25, Sigma Aldrich, USA) and washed commercial silica granules (average size 1±0.3mm, obtained from local market in Minya, Egypt) were hydrothermally treated in a Teflon-lined hydrothermal reactor (200 ml volume) at different temperatures (140, 150, 170 and 185°C) for 10 h. The reaction time has been selected based on previous studies to ensure maximum reaction yield. For instance, Tai Thien Huynh et al. have reported very good results when the hydrothermal treatment has been done at 8.0 h [35, 36]. Based on $TiO_2$: sand weight ratio, different samples were prepared; 1:5, 1:10 and 1:20. The filtered granules were washed and calcined at 700°C for 2 h holding time. A 5 mm internal diameter and 12 cm height glass tube was filled with the functionalized silica granules and utilized as a packed bed for dye removal (10 ppm) with a flow rate of 3.53 ml/min under 2000 W halogen lamp. 0.5 g of the prepared catalyst was used in hydrogen generation experiment using 100 ml of a scavenger-free domestic wastewater under direct sunlight in September from 11:00 to 14:00. The granules have been separated and reused in a successive run. In these experiments, the wastewater/granules suspension was placed in a well-sealed round-bottom flask with one opening from which a rubber pipe was fixed. Then the rubber pipe was immersed in a water-filled inverted graduated glass tube; the evolved gases were collected over the water surface. The accumulated gas consists mainly of $H_2$ and $O_2$ with a molar ratio of 2:1. The number of moles of accumulated hydrogen was calculated by recording the change in the volume above the water level using the following equation:

$$n = \frac{2 \times 273 \times V}{3 \times 22.4 \times T} \tag{1}$$

where $n$ is the number of moles of $H_2$ [mmol/g], $V$ is the volume of the gas (mL), and $T$ is the temperature of the solution (K). Then, a linear regression analysis for the collected data between the time and the number of moles of the collected hydrogen was done.

**Table 1. Chemical Characterizations of the utilized wastewater.**

| pH | COD[a] (mg/l) | BOD[b] (mg/l) | TSS[c] (mg/l) | TDS[d] (mg/l) | Total P[e] (mg/l) | VSS[f] (mg/l) | Total N[g] (mg/l) | Alk[h] (mg/l) |
|---|---|---|---|---|---|---|---|---|
| 7.45±0.03 | 305±15 | 269±5 | 65±5 | 554±17 | 3.594±0.01 | 155±3 | 4.3±0.15 | 235±7 |

[a] Chemical oxygen demand [b] Biological oxygen demand [c] Total suspended solid

[d] Total dissolved solid [e] Total phosphorous content [f] Volatile suspended solid

[g] Total nitrogen content [h] Total alkalinity

## 2.2 Characterizations

The characterizations have been performed in the Central lab for Microanalysis and Nano-technology, Minia University. The surface morphology of the functionalized silica particles was analyzed using a scanning electron microscope (SEM JSM-IT200, JEOL, Japan) equipped with EDX analyzer. X-ray diffractometer from Rigaku (XRD, Japan) was utilized to investigate the chemical composition of the modified silica after the calcination process. High resolution optical microscope with DP73 Digital Camera, Olympus LS was used to get 2D and 3D high resolution images. In El-Minya governorate, Egypt, there is a long (135 Km) water drain (MASRAF Al-MOHEET (, receiving around 9000 $m^3$/day industrial, municipal and agriculture wastewaters. For water photo splitting process, samples were collected from this water drain and characterized in Sanitation and Drinking Water Company Labs, El-Minya, Egypt. The samples were withdrawn from the drain during August and September from several locations. Characteristics of the used water are summarized in Table 1.

## 3. Results and discussion

### 3.1 Characterization of the produced catalyst

The dissolution of $TiO_2$ nanoparticles (NPs) refers to the process of breaking down the solid NPs into smaller particles or ions in a liquid solution. The rate of dissolution is influenced by several factors including surface area, size, and structure of the NPs, as well as the properties of the solvent. It was proved that, during the hydrothermal treatment, $TiO_2$ NPs can be broken down to $H_2Ti_3O_7.xH_2O$ nanocrystals which can be adsorbed at the surface of the used silica granules. In other words, the subcritical water does have the capability to convert the chemical structure of this important semiconductor to be an absorbable species. Later on, during the cooling process $H_2Ti_3O_7.xH_2O$ nanocrystals are hydrolyzed to $TiO_2$ thin layer covering the supporting material through a sequence of transformation series [37, 38]:

$$H_2Ti_3O_7xH_2O \rightarrow H_2Ti_3O_7 + xH_2O \rightarrow 3TiO_2 + (x+1)H_2O \qquad (2)$$

Therefore, the preparation process is carried out in different steps which were graphically illustrated in Fig 1. Briefly, due the new properties of the subcritical water, the R25 NPs are breaking down to $H_2Ti_3O_7.xH_2O$ nanocrystals which could be adsorbed on the surface of silica granules. Upon cooling, the formed hydrogen titanate is hydrolyzed (Eq 2) into titanium oxide NPs. However, to increase the adhesion force between the adsorbed $TiO_2$ crystals and the sand granules, calcination process is required.

Fig 2A displays optical electronic microscope image for the used pristine sand molecules. As shown, the surface is almost clean. Fig 2B represents 3D image for the pristine surface. It is clear that the surface is rough as the image shows bumps and ridges on the surface. This irregular texture provides high surface area for adsorption of function materials. Fig 2C demonstrates SEM image for the used $TiO_2$ nanoparticles. It can be concluded that the average diameter of this metal oxide particles is about 7±0.3 μm.

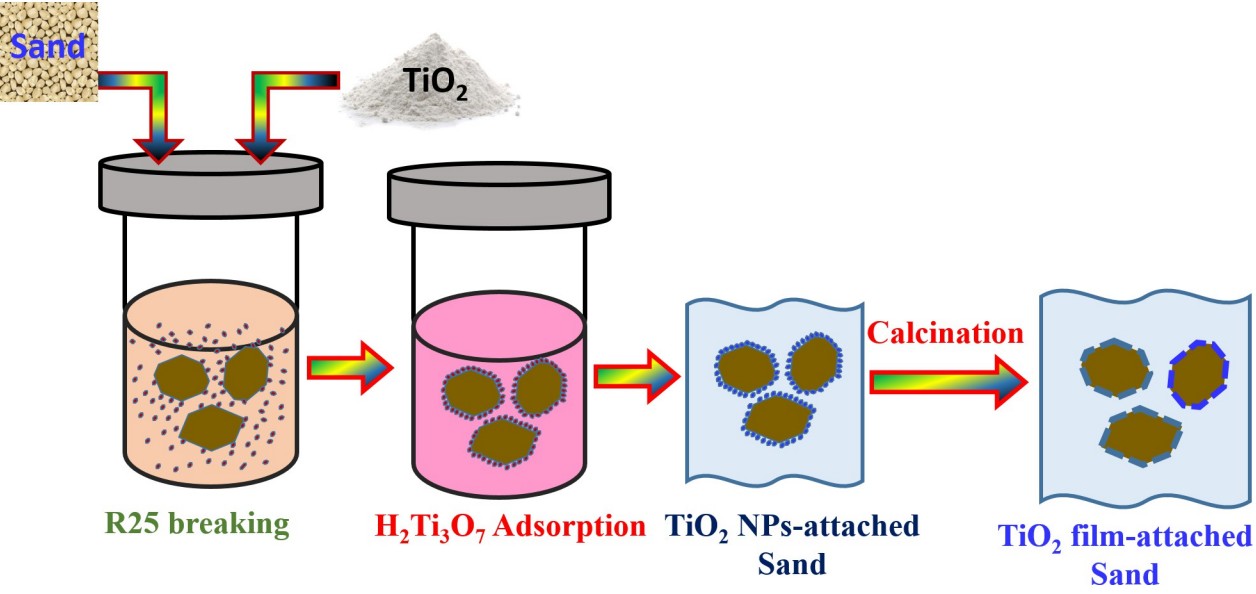

**Fig 1. Schematic diagram showing the preparation steps of the proposed TiO₂-deposited silica granules.**

The electronic microscope images of the produced coated sand granules, prepared at 170˚C hydrothermal temperature using TiO₂:sand ratio of 1:10 and calcined at 700˚C, are displayed in Fig 3. As shown in Fig 3A and 3B, the sand surface was decorated by large and small white

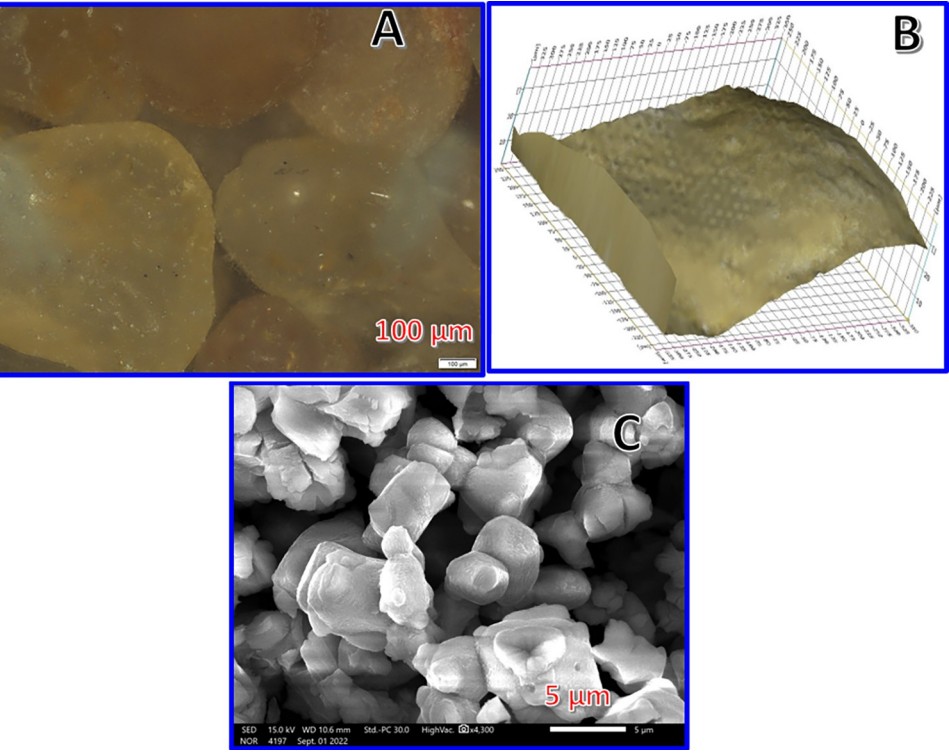

**Fig 2.** High resolution electronic optical microscope for the used sand; (A), 3D image of the pristine sand surface; (B) and SEM image of the used titanium oxide particles; (C).

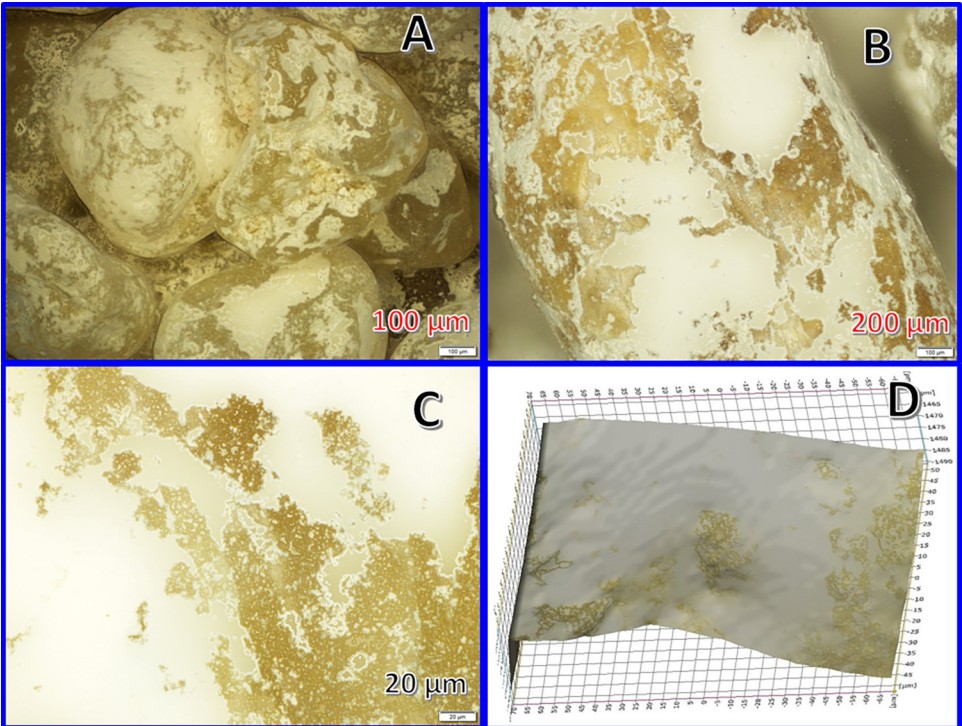

**Fig 3.** High resolution electronic optical microscope image for the prepared TiO$_2$-coated sand granules; (A), two magnifications images for the coated granules surface; (B) and (C), and 3D image for the surface; (D). The investigated sample was prepared at 170 and 700°C hydrothermal and calcination temperature, respectively.

spots. The high resolution image in Fig 3C concludes that these spots have smooth and cracked-free surfaces. This finding was further supported by the 3D image (Fig 3D), it can be seen that the white spots are thin and tightly stuck with the sand surface. Considering the large particle size of the used titanium oxide particles (Fig 2C), it can be claimed that formation these white spots were achieved due to an extreme deformation of TiO$_2$ particles. In other words, formation of these continuous thin white spots was a result of adsorption of very small crystals and/or large molecules. Accordingly, it is highly expected that the relatively large used TiO$_2$ particles were partially dissolved during the hydrothermal treatment process which supports the aforementioned proposed mechanism.

Fig 4 displays Energy Dispersive X-Ray (EDX) analysis for TiO$_2$-deposited silica for a sample prepared from 1:20 TiO$_2$: sand ratio at hydrothermal treatment of 170°C and sintered at 700°C. As shown, titanium could be detected on the surface with high content even compared to silicon. Numerically, the surface of the modified silica contains titanium, silicon and oxygen with atomic percentage of 19.58, 3.49 and 76.49%, respectively. This finding was expected due to the observed surface of the treated silica granules; Fig 3.

Fig 5 displays the X-ray diffraction (XRD) patterns for three samples, as-obtained silica granules after the hydrothermal treatment process in presence of TiO$_2$ R25 at 170°C, and after calcination the same sample at 500 and 700°C. Sand cannot be considered a single compound. Indeed, quartz (SiO$_2$) represents the major constituent (93–95 wt.%), there are several other oxides are incorporated in the natural sand such as iron oxides (FeO, Fe$_2$O$_3$), aluminum oxide, lime (CaO), soda (Na$_2$O), potash (Kro)... etc. [39, 40]. On the other hand, the used TiO$_2$ R25 composes of a single constituent; rutile phase titanium oxide. Accordingly, the observed diffraction peaks at 2θ values of 25.45°, 36.24°, 41.27°, 54.35°, 56.65°, 62.23° and 69.12°

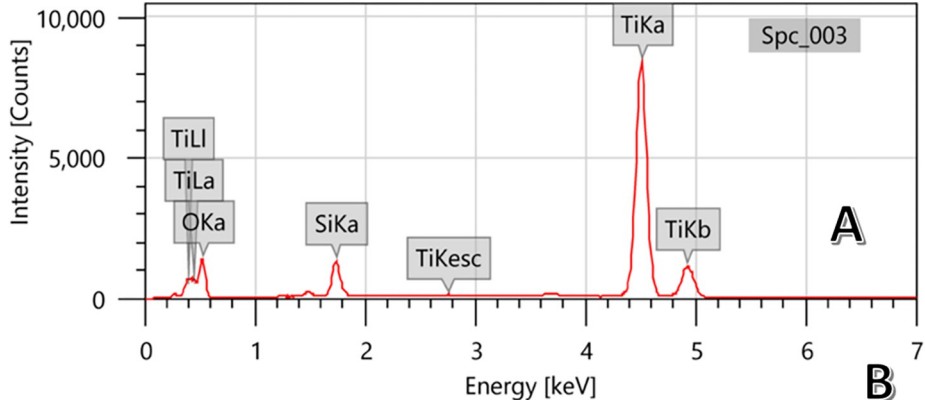

| Element | Line | Mass% | Atom% |
|---|---|---|---|
| O | K | 54.31±0.62 | 76.94±0.88 |
| Si | K | 4.32±0.06 | 3.49±0.05 |
| Ti | K | 41.37±0.17 | 19.58±0.08 |
| Total | | 100.00 | 100.00 |
| Spc_003 | | | Fitting ratio 0.0126 |

**Fig 4.** EDX spectrum; (A) and the mass and atomic analyses; B. The sample was prepared at 170˚C hydrothermal treatment temperature and calcined at 700˚C.

corresponding to the crystal planes (110), (101), (111), (211), (220), (002) and (301), respectively represent the rutile phase in the investigated samples [JCPDS #21–1276]. It is noteworthy mentioning that most of the diffraction peaks of quartz and rutile are close and almost overlapped. For the hydrothermally treated sample, the highest peak in the pattern at two theta angle of ~ 60.1˚ corresponding to $d$ spacing of 0.152 nm can be assigned to (211) crystal plane in the quartz ($SiO_2$) crystal according to ICDD #46–1045 [41]. This peak can be considered a fingerprint for the quartz in the sample as there is no a close rutile peak to this peak. Therefore, decreasing the intensity of this peak in the pattern of the sintered samples can be explained as change in the crystal structure of the quartz during the sintering process and/or covering the silica granules by a highly crystalline layer. Practically, the quartz crystal can be subjected to little deformation upon heating, however this deformation does not lead to destroying a high density crystal plane [42]. Therefore, it can be claimed that the very low decrease in the intensity of this peak, in the patterns representing the sintered samples, is mainly attributed to coating of the sand granules by a highly crystalline layer; $TiO_2$. On the other hand, the main peak of rutile phase at 2theat value of 27.4˚ corresponding to (110) crystal plane and $d$ spacing of 0.325 nm is close to quartz crystal diffraction peak at 26.6˚ ($d$ = 0.334 and (101) crystal plane) which resulted in merging these two peaks in the obtained patterns. It is clear that the resultant peak of merging these two standard peaks became predominant after the calcination process; Fig 5. This finding reveals that the crystallinity of the deposited $TiO_2$ layer was enhanced due to applying the thermal treatment process. In details, XRD results further support the aforementioned proposed mechanism. Typically, as it was explained in the proposed mechanism that, during the hydrothermal treatment process, the used $TiO_2$ particles were dissolute to form $H_2Ti_3O_7$ nanocrystals which were adsorbed on the silica surface. Consequently, for the thermally untreated sample, the intensities of the rutile representative peaks were very small. However, after the calcination process, rutile in the sample was the main phase and its peaks even overlaid the quartz peaks. Therefore, it is safe to claim that, during the calcination

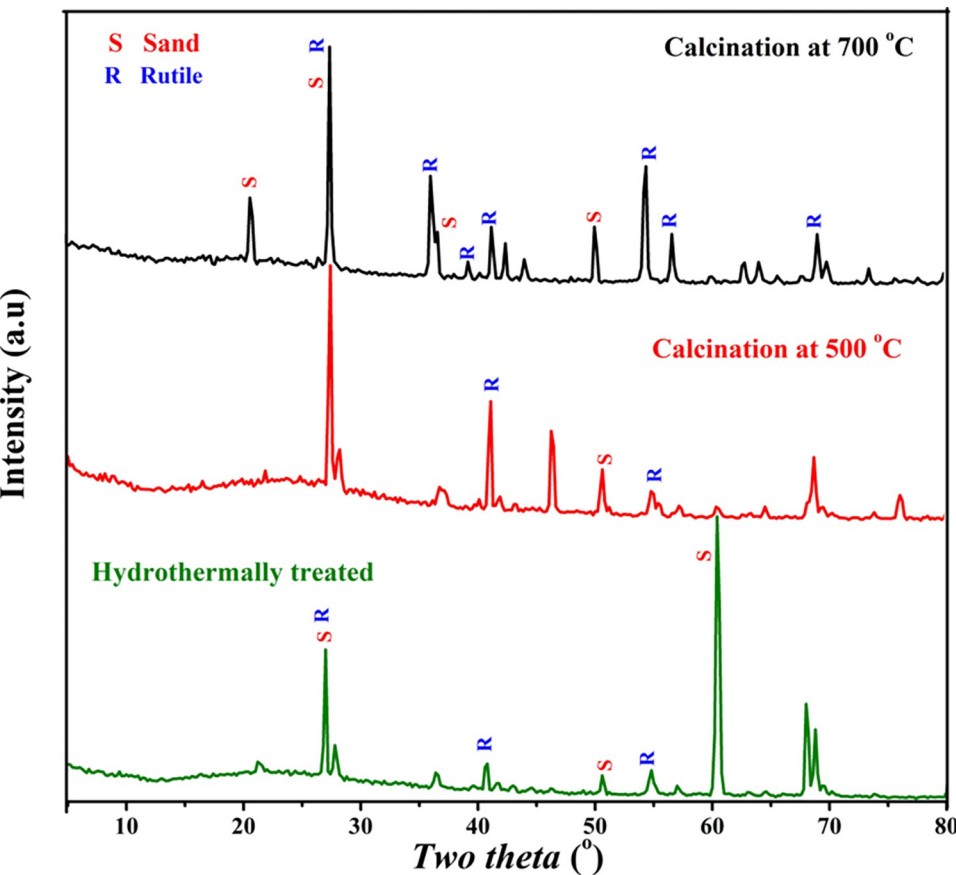

**Fig 5. XRD patterns for the hydrothermally treated (at 170˚C, without calcination), and after calcination at 500 and 700˚C functionalized silica granules.**

process, the adsorbed $H_2Ti_3O_7$ nanocrystals were thermally decomposed to $TiO_2$ thin and dense layer attaching the sand surface. Moreover, increasing the calcination temperature from 500 to 700˚C led to improve the crystallinity of the deposited layer as can be concluded from increasing the intensity of the rutile peaks.

The band gap energy of a material is a crucial factor that determines its photo catalytic activity. In photo catalysis, a material absorbs light energy to excite electrons from the valence band to the conduction band, creating electron-hole pairs that can participate in chemical reactions. The energy required to excite electrons from the valence band to the conduction band is determined by the band gap energy of the material. A material with a narrow band gap energy absorbs light of lower energy, while a material with a wide band gap energy absorbs light of higher energy [43, 44].

The band gap energy of a material can be determined by different ways. It can be estimated from its UV-Vis absorption spectrum. In the UV-Vis spectrum, the absorption edge corresponds to the energy required to excite electrons from the valence band to the conduction band [43]. Moreover, the band gap energy can be estimated from a plot of photon energy versus absorption coefficient ($\alpha$) or absorption wavelength ($\lambda$) using the Tauc plot method. The Tauc plot is a linear regression analysis of the absorption coefficient ($\alpha$) as a function of photon energy (hv) in the vicinity of the absorption edge [45]. Tauc proposed utilizing optical absorption spectra to estimate the band gap energy of amorphous semiconductors[46]. Davis and Mott explored his concept further[47]. This method has been invoked to estimate the band

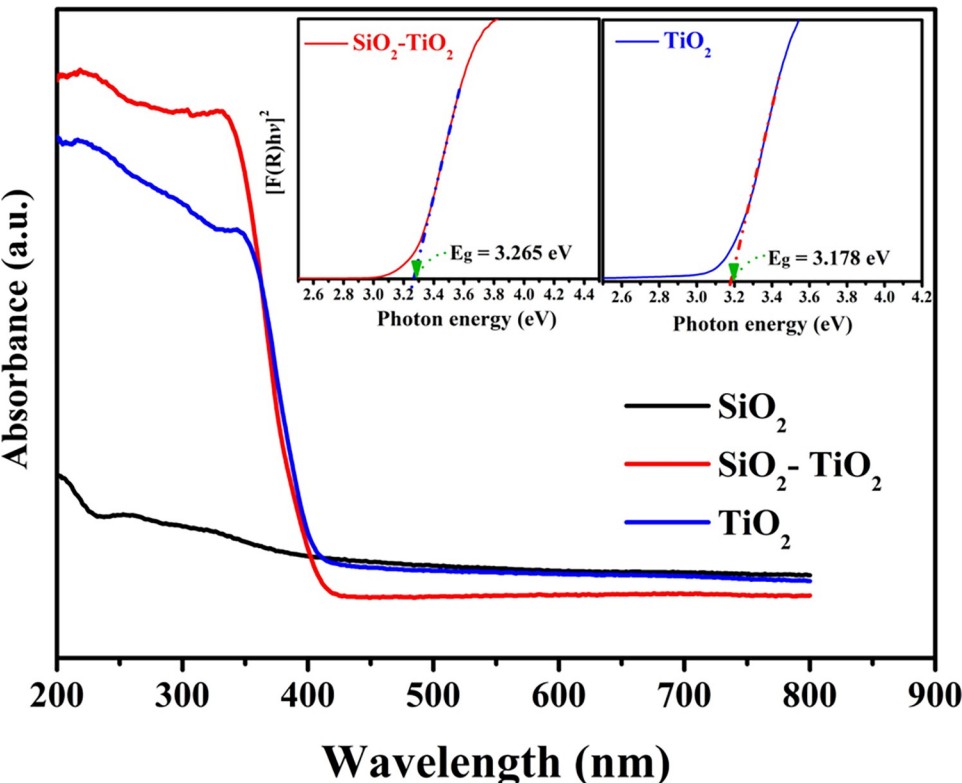

**Fig 6. UV-vis. spectra for TiO$_2$, SiO$_2$ and SiO$_2$-TiO$_2$ composite.** The insets represent Tauc plots for determination the band gap energies for the pristine TiO$_2$ and SiO$_2$-TiO$_2$ composite.

gap of TiO$_2$ and TiO$_2$-SiO$_2$ as shown in the insets of Fig 6 which displays the UV-vis spectra for pristine TiO$_2$ and SiO$_2$, and their composite. Numerically, the band energies for the pristine and modified TiO$_2$ nanofibers are 3.187 and 3.265 eV, respectively. This finding can conclude both formulations are more efficient under UV radiation. This hypothesis might be valid for the pure semiconductor. However, in case of composite semiconductor, the interaction between the constituents' band gaps can result in generating high photocatalytic activity under visible radiation [44, 48]. This supposition was experimentally supported as it will be shown in this study.

## 3.2 Water photo splitting

In a photo splitting reaction, light energy is used to separate water into its constituent parts: hydrogen and oxygen. However, the reaction can produce excess electron holes or electrons that can quickly recombine and reduce the efficiency of the reaction [34]. Scavengers are substances that are added to the reaction mixture to remove these excess electron holes or electrons and prevent them from recombining [49]. Moreover, the scavenger can prevent the formation of reactive oxygen species (ROS) such as hydrogen peroxide (H$_2$O$_2$) and superoxide (O$^{-2}$). Other scavengers can act as electron donors instead of water to neutralize the excess charge and prevent recombination [50]. Overall, scavengers play a crucial role in ensuring the success and efficiency of water photo splitting reactions. However, from the economical point of view, scavenger-less water photo splitting process is highly preferable as it will save the scavenger cost.

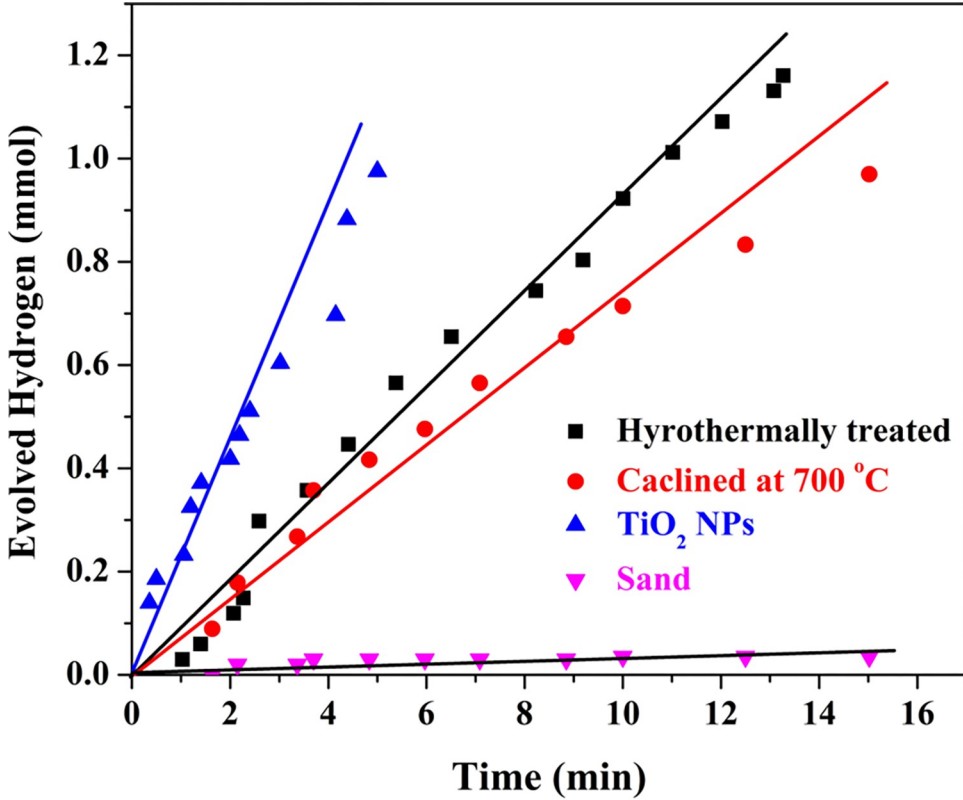

**Fig 7. Hydrogen generation rate under direct sunlight radiation for the raw and calcined hydrothermally treated sand granules.** For comparison, hydrogen generation rate was investigated using $TiO_2$ NPs and naked sand granules under similar conditions.

Some organic pollutants can be exploited as photo sensitizers for a scavenger-free water photo splitting reaction under visible light [45]. These pollutants absorb light and generate electron-hole pairs that promote the reduction and oxidation reactions of water. The use of organic pollutants as photo-sensitizers makes the process environmentally friendly, as these pollutants are typically present in wastewater and can be recovered and used instead of being disposed of. The aforementioned hypotheses were proved by investigation the photocatalytic activity of the proposed material toward water splitting using domestic wastewater as it contains several kinds of organic pollutants. Typically, the sewage wastewater can contain saccharides, amino acids, fatty acids, hydroxyacids, aromatic compounds and steroids [51]. Therefore, the analysis of the utilized sewage wastewater, as indicated in Table 1, reveals high COD and BOD contents. Moreover, there are considerable amount of phosphorous and nitrogen –containing compounds. Fig 7 shows the hydrogen generation rate, in a scavenger-less wastewater solution, using the hydrothermally treated modified silica granules and after the calcination process under the direct sunlight. The investigated samples were prepared at 170 and 700°C hydrothermal and calcination temperatures, respectively and 1:20 $TiO_2$:sand ratio. Moreover, for comparison, the photo catalytic activity of the used $TiO_2$ nanoparticles and bared silica granules toward water photosplitting was investigated as well. It is noteworthy mentioning that gas analysis indicates that the obtained gas is $CO_2$-free which assures that the collected gas was generated from water splitting. As shown in the figure, a hydrogen production rate of $93 \times 10^{-3}$ mmol/s could be achieved from the hydrothermally treated $TiO_2$–deposited sand granules. Applying the calcination process has a negative impact on the

photocatalytic activity as can be observed from the figure. Typically, the hydrogen production rate has been decreased to be ~ $75 \times 10^{-3}$ mmol/s. This finding can be attributed releasing of some $TiO_2$ NPs from the thermally untreated sand granules surface due the weak adhesion forces. 25 mg from $TiO_2$ nanoparticles was used during checking the photocatalytic activity of the utilized precursor which almost matches the used amount during preparation of the investigated amount from the functionalized sand granules. As shown, the free $TiO_2$ NPs have the highest photo catalytic activity among the investigated samples; $227 \times 10^{-3}$ mmol/s hydrogen production rate was observed. Finally, the uncovered sand granules have shown trivial photo catalytic activity toward water splitting reaction. Certainly, the surface area of the titanium oxide nanoparticles has decreased due to the proposed fixation strategy, however the influence on the photo catalytic activity was not upsetting. This result might be assigned to the thinness of deposited layer and/or the role of $SiO_2$ in enhancing the photo catalytic activity of titanium oxide [52].

The reusability of a photocatalyst is an important factor to consider in practical applications. It refers to the ability of the photocatalyst to maintain its photocatalytic activity after multiple cycles of use [53, 54]. In the hydrogen production experiments, the photo catalyst granules/domestic wastewater slurry was stirred at at 500 rpm. Therefore, if the adhesion force between the adsorbed $TiO_2$ nanoparticles and the surface of the sand molecules is low, the $TiO_2$ nanoparticles will be detached during the mixing process. Indeed, the hydrothermal treatment could be successfully utilized to coat the sand granules by a thin layer from $TiO_2$ nanoparticles, however strong adhesion is not guaranteed. Therefore, the calcination process was carried out to make a sintering between the two oxides. To scientifcially prove this hypothesis, the calcined granules were used in two successive runs; Fig 8. After the first run, the

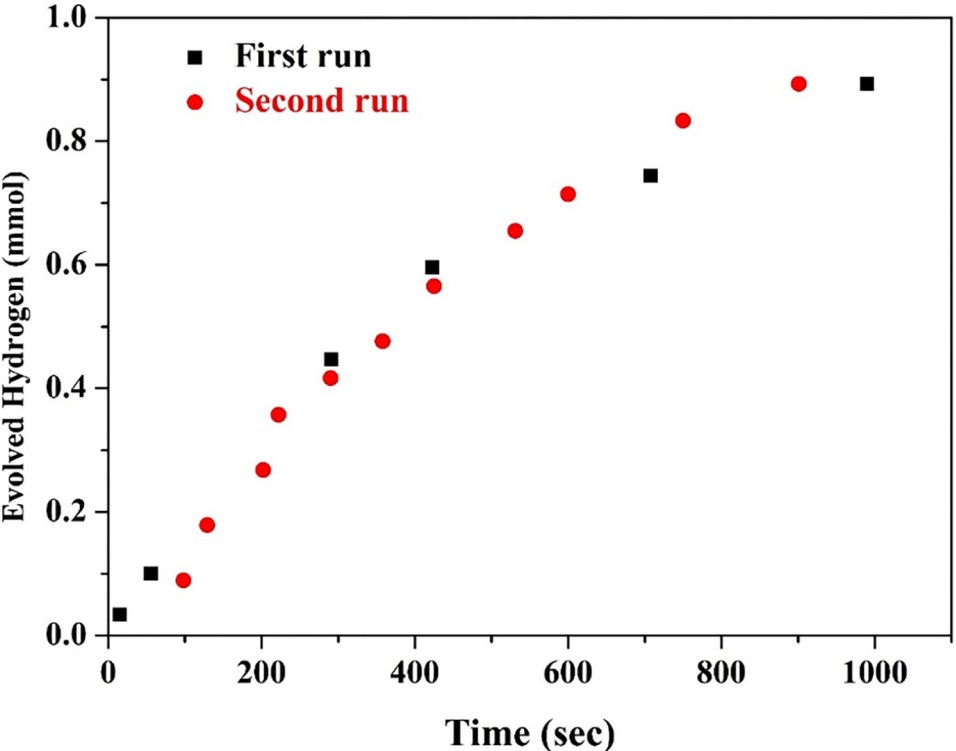

**Fig 8. Hydrogen generation rate under direct sunlight radiation for two successive cycles using functionalized silica prepared at 170˚C hydrothermal treatment temperature and calcined at 700˚C.**

granules were separated, washed and used in the second run. It can be claimed that obtaining almost similar performance in two successive cycles confirms strong stability of the modified silica granules. This conclusion is also supported by the relatively higher photocatalytic activity of the thermally untreated samples (Fig 7) which was explained as release of some attached $TiO_2$ nanoparticles due to the low adhesion force with the surface of the sand granules.

Influnce of the hydrothermal treatment temperature on the photocatalytic activity toward water splitting reaction was investigated. The proposed $TiO_2$-coated sand granules were prepared at different temperatures; 130, 150, 170 and 185˚C. The results are displayed in Fig 9. The results indicated that increasing the hydrothermal temperature enhances the photo catalytic activity in the form of increasing the hydrogen production rate. However, the improvement is not high which concludes that the dissolution of $TiO_2$ nanoparticles in the subcritical water can be carried out at relatively low temperature.

Table 2 summarizes the hydrogen production rate for some recently reported photo catalysts using chemicals as scavangers. As shown in the table, the proposed catalyst shows good activity comparatively. Beside, the proposed catalsyst possesses two main advanatages; the catalyst is fixed on the surface of large granules, and organic pollutants were exploited as scavanger.

Based on previous reports, the good photocatalytic activity of the proposed composite can be attributed to the role of silica in remedation of the fast electrons/holes recombination problem [64, 65]. Silica can act as an electron acceptor, promoting electron transfer from $TiO_2$ to the $SiO_2$ surface. The $SiO_2$ layer on the $TiO_2$ surface also serves as a barrier that prevents electron-hole recombination, leading to longer charge carrier lifetimes and improved

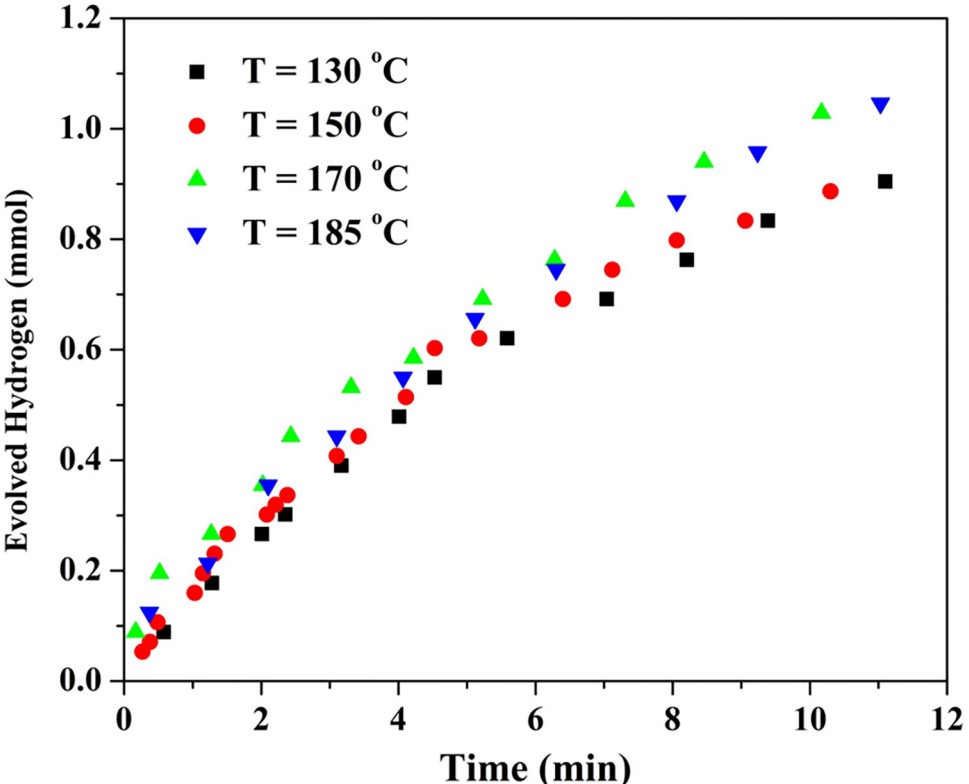

**Fig 9. Effect of the hydrothermal treatment temperature on the hydrogen production rate using functionalized silica prepared from 1:20 $TiO_2$:Sand wt. ratio.**

**Table 2. A comparison of the hydrogen evolution rate for different nanocatalysts.**

| Photocatalyst | Scavenger agent | H$_2$ production (mmol H$_2$/gcat. Min) | Ref. |
|---|---|---|---|
| Pt/ TiO$_2$ nanosheet | Ethanol | 0.0056 | [55] |
| Graphene/TiO$_2$ NPs | Methanol | 0.0123 | [56] |
| TiO$_2$ NPs | Methanol | 0.1 | [57] |
| (Pt/HS-TiO$_2$) | Methanol | 0.017 | [58] |
| Pt-doped TiO2–ZnO | Methanol | 0.0034 | [59] |
| Pt-TiO2 particles | Methanol | 0.444 | [54] |
| Cd-TiO$_2$ nanotube | Methanol | 24 | [33] |
| CdS/TiO$_2$ mesoporous core-shell | Na$_2$S/ Na$_2$SO$_3$ | 1.13 | [60] |
| Ni/TiO2 nanotube | - | 0.433 | [61] |
| Ni/GO-TiO$_2$ nanoparticles | Methanol | 3 | [62] |
| Ag-TiO$_2$ NFs | Na$_2$S/ Na$_2$SO$_3$ | 2 | [34] |
| NiCo$_2$S$_4$/CdO@CC | - | 0.00125 | [63] |
| Cd-doped TiO$_2$ NPs | Na$_2$S/ | 0.7 | [44] |
| Cd-doped TiO$_2$ NFs | Na$_2$SO$_3$ | 16.5 | [44] |
| TiO$_2$/coated sand | Organic pollutants | 3* | This study |

* This value was estimated based on amount of TiO$_2$ in the used sample

photocatalytic efficiency. For isnatnce, deposition of SiO$_2$ on TiO$_2$ nanotube arrays improved the efficiency of hydrogen production from water splitting. The SiO$_2$ layer was found to improve the charge transfer properties of TiO$_2$ by reducing the recombination rate of photo-generated electrons and holes [66]. Moreover, SiO$_2$ layer on the surface of TiO$_2$ nanotubes improved the photocatalytic activity for the degradation of methylene blue under visible light irradiation. The SiO$_2$ layer was found to act as a barrier that prevented electron-hole recombination and improved the charge transfer properties of TiO$_2$ [67].

## 3.3 Continuous mode dye photo degradation

A packed bed dye photodegradation system consists of a packed bed filled with a photocatalytic material that is exposed to light energy. The *S* curve is a common plot used to describe the continuous adsorption process. It shows the relationship between the breakthrough time and the adsorbate concentration in the effluent stream. At the beginning of the adsorption process, the concentration of the adsorbate in the effluent stream is low, and the adsorbent bed is able to effectively remove the adsorbate. As the adsorption process continues, the concentration of the adsorbate in the effluent stream increases, and eventually reaches a point where the adsorbent bed becomes saturated and is no longer able to effectively remove the adsorbate. This point is known as the breakthrough point. The *S* curve shows a gradual increase in the effluent concentration of the adsorbate until the breakthrough point is reached, at which point the concentration rapidly increases. The shape of the curve resembles an "S", with a gradual increase in the early stages followed by a steep increase as the breakthrough point is reached. Fig 10 demonstrates the breakthrough (*S* shape) curves for methylene blue dye removal using a packed bed from the modified silica prepared at different TiO$_2$: sand ratios at a flow rate of 3.53 ml/min. In contrast to the normal adsorption process, theoretically, in the photodegradation removal of dye, the used catalyst is not exhausted with time. Instead, the removal effecinecy is going constantly with time. However, in the packed bed, the dye degrdation products are not removed fastly from the reaction media especially at low stream flow rate. Moreover, some by products might have adsorption affinity on the catalyst surface. Consequently, the

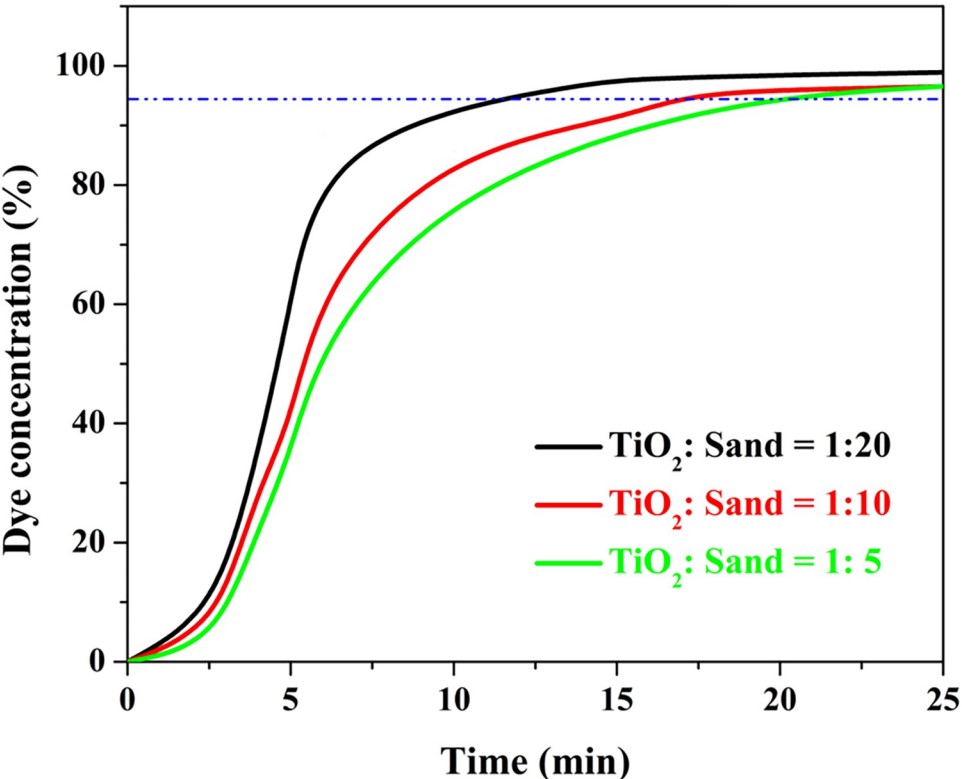

**Fig 10. Breakthrough curve using the proposed activated silica prepared from different TiO₂:Sand ratios at hydrothermal treatment temperature of 170˚C.**

photodegrdation efficeincy decreases with time and also give breakthrough plot [68]. Considering ~ 95% removal effeciency is an acceptable theresholed, it is clear that the exhaustion point, the elapsed time that correspondig to the desired removal percentage, is strongly affected by $TiO_2$: Sand weight ratio. In details, as shown in Fig 10, the exhaustion point was 12.3, 17.4 and 21.3 min for the modified sand granules prepared from a mixture having $TiO_2$: sand ratio of 1:20, 1:10 and 1:5, respectively. This finding is acceptable as increasing $TiO_2$ content leads to enlarge the deposited active spots on the sand surface.

Influence of the thermal treatment step was also investigated in case of utilizing the proposed coated sand granules in a continuous dye degradation process. Fig 11 demonstrates the breakthrough curves dye removal process using a packed bed from $TiO_2$ –coated sand granules prepared at 170˚C using a slurry contains 20% $TiO_2$ NPs with respect to sand particles before and after calcination at 700˚C. It is clear from the results that the exhaustion point increased distinctly which indicates high photo catalytic activity for the sintered silica compared to the thermally untreated ones.

As aforementioned, when a photon is absorbed by the $TiO_2$, it can create an electron-hole pair. The electron can move to the conduction band of the material, while the hole moves to the valence band. However, if the electron and hole recombine before they can participate in a chemical reaction, the energy from the photon will be wasted and the photocatalytic efficiency of the material will be reduced. An effective approach is to modify the surface of the $TiO_2$ material with co-catalysts or sensitizers. Co-catalysts can help to facilitate the transfer of electrons or holes to the surface of the material, while sensitizers can absorb photons and transfer

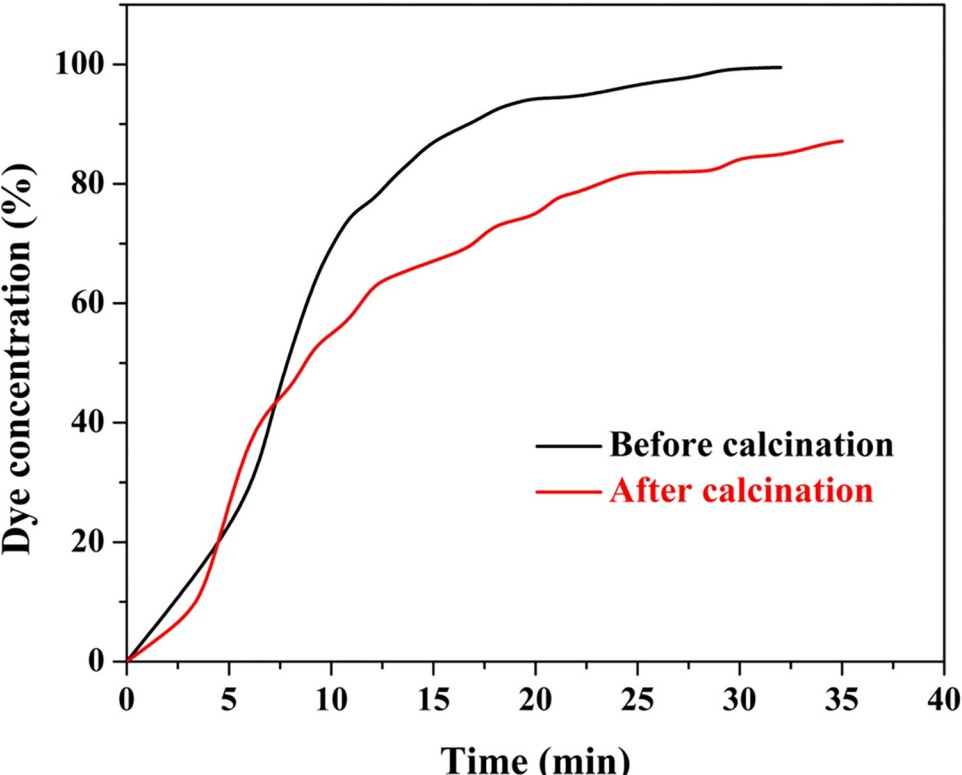

**Fig 11. Breakthrough curve for dye removal using raw and thermally treated TiO₂-coated silica granules prepared at hydrothermal treatment temperature of 170˚C and calcined at 700˚C.**

the excited electron to the TiO$_2$ material, reducing the recombination rate of the electron-hole pairs [45, 69]. In this regard, SiO$_2$ showed good activity as a co-catalyst [69]. Therefore, the higher photo catalytic activity of the sintered sample might be attributed to good adhesion of the titanium oxide nanoparticles on the surface of the sand granules which eleminates the interfacial resistances between the two oxides that enhances the role of SiO$_2$ in remedation of the electrons/holes recombination problem. Finally, it is noteworthy mentioning that the chosen flow rate was selected to be consistent with the cross section area of the used bed. Numerically, the selected flow rate matches around 180 liter/m$^2$.min which is considered high flux rate and highly accepted from the industrial point of view. However, it is planned to study the effect of flow rate in a separate study.

## 4. Conclusions

Hydrothermal treatment of TiO$_2$ NPs/silica granules mixture at high temperature results in deposition the photo catalyst NPs on the surface of the used support. The hydrothermal treatment temperature relatively affects the performance of the proposed catalyst; 170˚C is the best value. Calcination is required to enhance the adhesion force. The modified silica can be utilized as a packed bed for continuous removal of the organic pollutants-containing wastewaters. The exhaustion point in the dye removal breakthrough curves strongly increases in case of the sintered sample due to overcoming the electrons/holes recombination problem. Moreover, the introduced activated silica is a good and stable photo catalyst for water splitting from a scavenger-less wastewater medium under visible light.

## Acknowledgments

The authors would like to thank the Deanship of Scientific Research, Qasim University, for funding the publication of this paper.

## Author Contributions

**Funding acquisition:** Osama M. Irfan.

**Investigation:** Osama M. Irfan.

**Methodology:** Olfat A. Mohamed.

**Project administration:** Nasser A. M. Barakat.

**Validation:** Olfat A. Mohamed.

**Writing – original draft:** Nasser A. M. Barakat.

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
