## [Decision Letter · Decision Letter 0]

27 Apr 2023

PONE-D-23-10591TiO2 NPs-immobilized Silica Granules: New Insight for Nano Catalyst Fixation for Hydrogen Generation and Sustained Wastewater TreatmentPLOS ONE

Dear Dr. Barakat,

Thank you for submitting your manuscript to PLOS ONE. After careful consideration, we feel that it has merit but does not fully meet PLOS ONE’s publication criteria as it currently stands. Therefore, we invite you to submit a revised version of the manuscript that addresses the points raised during the review process.

We look forward to receiving your revised manuscript.

Kind regards,

Van-Huy Nguyen, Ph.D.

Academic Editor

PLOS ONE

Journal Requirements:

"The authors would like to thank the Deanship of Scientific Research, Qasim Uni-versity, for funding the publication of this paper."

5. We note that Figures 1, 2 and 3 in your submission contain copyrighted images. All PLOS content is published under the Creative Commons Attribution License (CC BY 4.0), which means that the manuscript, images, and Supporting Information files will be freely available online, and any third party is permitted to access, download, copy, distribute, and use these materials in any way, even commercially, with proper attribution. For more information, see our copyright guidelines: http://journals.plos.org/plosone/s/licenses-and-copyright.

a. You may seek permission from the original copyright holder of Figures 1, 2 and 3 to publish the content specifically under the CC BY 4.0 license. 

Reviewers' comments:

Reviewer's Responses to Questions

**Comments to the Author**

1. Is the manuscript technically sound, and do the data support the conclusions?

Reviewer #1: Yes

Reviewer #2: Yes

Reviewer #3: Yes

2. Has the statistical analysis been performed appropriately and rigorously? 

Reviewer #1: Yes

Reviewer #2: Yes

Reviewer #3: N/A

3. Have the authors made all data underlying the findings in their manuscript fully available?

Reviewer #1: Yes

Reviewer #2: Yes

Reviewer #3: Yes

4. Is the manuscript presented in an intelligible fashion and written in standard English?

Reviewer #1: Yes

Reviewer #2: Yes

Reviewer #3: No

5. Review Comments to the Author

Reviewer #1: Manuscript ID: PONE-D-23-10591

Title: TiO2 NPs-immobilized Silica Granules: New Insight for Nano Catalyst Fixation for

Hydrogen Generation and Sustained Wastewater Treatment

In this manuscript, the authors desribe a novel method for immobilizing R25 nanoparticles on silica granules using hydrothermal treatment and calcination was introduced for heterogeneous catalytic processes. The resulting composite showed good performance in removing methylene blue dye and improved photocatalytic activity for hydrogen generation from sewage wastewaters under direct sunlight. The topic of the paper is interesting but the manuscript contains information and some propositions that are not well-justified. Furthermore, additional experiments are required to evaluate the catalytic performance of the as-prepared material. Therefore, the paper cannot be accepted for publication in its present form and major revision is needed. Some suggestions were listed as follows:

1. Abstract and introduction sections should be concisely rewritten to emphasize and highlight the results and urgency of this work.

2. In this work, the authors used a hydrothermal process for fabricating materials, so the hydrothermal method should be introduced, and some previous studies (such as https://doi.org/10.1016/j.ijhydene.2018.09.174, https://doi.org/10.1166/jnn.2018.15719) should be mentioned as examples for advantages of the hydrothermal route.

3. The authors investigated the effect of temperature (140, 150, 170, and 185 oC) on the formations of the catalysts by the hydrothermal routes, so the authors should be showed the change of behaviors of catalysts according to the temperatures. Why the authors did not investigate the effect of reaction time, which is a vital factor in fabricating the materials?

4. The particle size of TiO2 before and after also should be confirmed by its effect on the catalytic performance. In addition, can the author control the TiO2 structure?

5. In photocatalysis applications, the band gap, and surface area are important parameters to improve the photocatalytic efficiency, so the authors should be determined such values of the as-made catalysts.

6. The photocatalytic mechanism of the as-obtained photocatalysts should be mentioned. Some references such as https://doi.org/10.1021/acs.jpcc.0c03590;
https://doi.org/10.1002/ceat.202200388 should be mentioned when discussing the photocatalytic mechanism.

7. Did the authors investigate the reusability of the catalyst for the photocatalytic degradation of dyes?

8. Some typos of English in the whole manuscript should be re-checked and improved.

Reviewer #2: The article entitled “TiO2 NPs-immobilized Silica Granules: New Insight for Nano Catalyst Fixation for Hydrogen Generation and Sustained Wastewater Treatment has significant scientific value.

In this work,. a novel approach for immobilizing R25 NPs on the surface of silica granules using hydrothermal treatment followed by calcination process. Due to the privileged characteristics of the subcritical water, during the hydrothermal treatment process, the utilized R25 NPs were partially dissolved and precipitated on the surface of the silica granules. Calcination at high temperature (700oC) resulted in improving the attachment forces. The structure of the newly proposed composite was approved by 2D and 3D optical microscope images, XRD and EDX analyses. The functionalized silica granules were used in the form of a packed bed for continuous removal of methylene blue dye. The results indicated that the TiO2:sand ratio has a considerable effect on the shape of the dye removal breakthrough curve as the exhaustion point, corresponding to ~ 95% removal, was 12.3, 17.4 and 21.3 min for 1:20, 1:10 and 1:5 metal oxides ratio, respectively. Moreover, due to inhibition of the electrons/holes recombination process, the calcination treatment distinctly improves the photo catalytic activity of the introduced coated silica. Furthermore, the modified silica granules could be exploited as a photocatalyst for hydrogen generation from sewage wastewaters under direct sunlight with a good rate; 75×10-3 mmol/s. Interestingly, after the ease separation of the used granules, the performance was not affected. Based on the obtained results, the 170 oC is the optimum hydrothermal treatment temperature . My comments listed below may help the authors further improve their work:

1. The authors can explain a bit about the adsorption effect using nanomaterials in the introduction part. Authors are suggested to enhance the discussion on adsorption kinetics and consider the following papers for the same:

https://doi.org/10.1016/j.matlet.2022.131716

https://doi.org/10.1016/j.surfin.2022.102182

https://doi.org/10.1007/s11356-022-20743-8

2. Authors need to improve the highlights as they are very generic and don’t exactly include the main points of their original work.

3. The novelty aspects of this research paper need to be further modified and compared with the already present research (if any) to further enhance the overall impact.

4. How the reusability of catalyst can be assured?. Please refer for some of the papers for a better explanation.

https://doi.org/10.1016/j.ijbiomac.2022.11.241

https://doi.org/10.1016/j.matlet.2022.131716

https://doi.org/10.1007/s11356-022-20743-8.

DOI: 10.1016/j.jmst.2021.01.051

5. The paper in its current state has so many typos, technical and formatting related errors. There are certain space, grammar, and English related errors in the manuscript which are significantly ignored. Authors are suggested to proofread the manuscript thoroughly and eliminate the errors such as subscripts, superscripts, uniformity in presentation (Fig.Xa&b/Fig.X a and b).

5. Have authors tested it under light just to see effect of photocatalysis

Reviewer #3: Please check typographical all the whole manuscripts. The author should rewrite the manuscript more concisely and succinctly, focusing on the obtained research results instead of rambling the textbook knowledge.

1. Introduction:

It should be shorter and more concise

2. Experimental part

It is not recommended to include basic knowledge from the textbook such as COD and BOD analysis procedures in the scientific article

3. The sections Results and discussion

Please focus on results and discussions in this section, do not include basic knowledge

There is no need to write too much about the concept, such as the concept of supercritical water.

It is not necessary to state the advantages and disadvantages and the purposes of the characterization of the produced catalyst analysis (EDX, XRD)

Fig 1 should be in section 2 Experimental part.

Table 1 should be used footnote for “Where COD is chemical oxygen demand, BOD is the biological oxygen demand, TSS is the total soluble suspended solids, TDS is the total dissolved solids, VSS is volatile suspended solids, total P is the total phosphorus, total N is the total nitrogen and Alk is the total alkalinity.” Paragraph.

Fig 4 should be added the legend

6. PLOS authors have the option to publish the peer review history of their article (what does this mean?). If published, this will include your full peer review and any attached files.

Reviewer #1: No

Reviewer #2: No

Reviewer #3: No

---

## [Author Response · Author response to Decision Letter 0]

25 May 2023

Dear Prof. Van-Huy Nguyen

Academic Editor

PLOS ONE

Thank you for your kind response about the manuscript [PONE-D-23-10591] entitled

 “TiO2 NPs-immobilized Silica Granules: New Insight for Nano Catalyst Fixation for Hydrogen Generation and Sustained Wastewater Treatment”

The referee's comments were helpful to strength the manuscript. We would like to inform you that we have modified the manuscript according to the newly given comments. 

To make it more easily, we have written the comments in bold phase followed by the responses in normal one. Moreover, in the revised manuscript, you can find the changes in the text in blue color. 

 We hope our responses cover all the comments. It will be our pleasure to respond about any more comments.

Thank you for your cooperation 

Sincerely yours 

Corresponding author

Nasser A. M. Barakat

Professor

Chemical engineering dep., Minia university, Egypt

Reviewer #1: Manuscript ID: PONE-D-23-10591

Title: TiO2 NPs-immobilized Silica Granules: New Insight for Nano Catalyst Fixation for

Hydrogen Generation and Sustained Wastewater Treatment:

In this manuscript, the authors describe a novel method for immobilizing R25 nanoparticles on silica granules using hydrothermal treatment and calcination was introduced for heterogeneous catalytic processes. The resulting composite showed good performance in removing methylene blue dye and improved photocatalytic activity for hydrogen generation from sewage wastewaters under direct sunlight. The topic of the paper is interesting but the manuscript contains information and some propositions that are not well-justified. Furthermore, additional experiments are required to evaluate the catalytic performance of the as-prepared material. 

Therefore, the paper cannot be accepted for publication in its present form and major revision is needed. Some suggestions were listed as follows:

1. Abstract and introduction sections should be concisely rewritten to emphasize and highlight the results and urgency of this work.

Response: Thank you for this comment, the abstract has been modified. 

2. In this work, the authors used a hydrothermal process for fabricating materials, so the hydrothermal method should be introduced, and some previous studies (such as https://doi.org/10.1016/j.ijhydene.2018.09.174, https://doi.org/10.1166/jnn.2018.15719) should be mentioned as examples for advantages of the hydrothermal route.

Response: Thank you for this comment, the given references containing helpful information to arise the advantages of the hydrothermal process; these references were cited in the introduction section (35 and 36).

3. The authors investigated the effect of temperature (140, 150, 170, and 185 oC) on the formations of the catalysts by the hydrothermal routes, so the authors should be showed the change of behaviors of catalysts according to the temperatures. Why the authors did not investigate the effect of reaction time, which is a vital factor in fabricating the materials?

Response: Thank you for this good comment. The reviewer is right; the hydrothermal reaction time is an important parameter. This study is introduced as a newly proposed methodology to exploit the hydrothermal process to immobilize TiO2 NPs on the surface of the sand granules. Therefore, after checking the literature, 10 h was selected as a proper reaction temperature. For instance, the authors in the suggested papers in the previous comment have reported good results when the hydrothermal treatment was assigned at 8.0 h. 

This explanation has been considered in the revised manuscript. 

4. The particle size of TiO2 before and after also should be confirmed by its effect on the catalytic performance. In addition, can the author control the TiO2 structure?

Response: Thank you for this comment. The used TiO2 NPs are commercial grade (R25 titanium oxide) and were used as received. As these nanoparticles are in nanoscale (as shown in Fig. 2C), we could not separate this material into different sizes to study this important parameter. However, this good comment gives a new idea for a future study by synthesizing titanium oxide nanoparticles in different sizes and study the influence of this important parameter. Moreover, the effect of morphology can be also studies; nanofibers, nanotubes and nanoparticles. 

5. In photocatalysis applications, the band gap, and surface area are important parameters to improve the photocatalytic efficiency, so the authors should be determined such values of the as-made catalysts.

Response: Thank you for this comment. The band gap has been determined in the revised manuscript; Figure 6 was newly added. 

6. The photocatalytic mechanism of the as-obtained photocatalysts should be mentioned. Some references such as https://doi.org/10.1021/acs.jpcc.0c03590;
https://doi.org/10.1002/ceat.202200388 should be mentioned when discussing the photocatalytic mechanism.

Response: Thank you for this comment. The given references were helpful to explain the mechanisms in the revised manuscript. 

7. Did the authors investigate the reusability of the catalyst for the photocatalytic degradation of dyes?

Response: Thank you for this comment. The reusability was checked in case of water photo splitting to investigate the stability of the proposed, so we did not check the reusability with the dye photo degradation after ensuring the good stability from water splitting reactions. 

8. Some typos of English in the whole manuscript should be re-checked and improved.

Response: Thank you for this comment. The whole manuscript was carefully revised. 

Reviewer #2: 

The article entitled “TiO2 NPs-immobilized Silica Granules: New Insight for Nano Catalyst Fixation for Hydrogen Generation and Sustained Wastewater Treatment has significant scientific value.

In this work,. a novel approach for immobilizing R25 NPs on the surface of silica granules using hydrothermal treatment followed by calcination process. Due to the privileged characteristics of the subcritical water, during the hydrothermal treatment process, the utilized R25 NPs were partially dissolved and precipitated on the surface of the silica granules. Calcination at high temperature (700oC) resulted in improving the attachment forces. The structure of the newly proposed composite was approved by 2D and 3D optical microscope images, XRD and EDX analyses. The functionalized silica granules were used in the form of a packed bed for continuous removal of methylene blue dye. The results indicated that the TiO2:sand ratio has a considerable effect on the shape of the dye removal breakthrough curve as the exhaustion point, corresponding to ~ 95% removal, was 12.3, 17.4 and 21.3 min for 1:20, 1:10 and 1:5 metal oxides ratio, respectively. Moreover, due to inhibition of the electrons/holes recombination process, the calcination treatment distinctly improves the photo catalytic activity of the introduced coated silica. Furthermore, the modified silica granules could be exploited as a photocatalyst for hydrogen generation from sewage wastewaters under direct sunlight with a good rate; 75×10-3 mmol/s. Interestingly, after the ease separation of the used granules, the performance was not affected. Based on the obtained results, the 170 oC is the optimum hydrothermal treatment temperature. 

My comments listed below may help the authors further improve their work:

1. The authors can explain a bit about the adsorption effect using nanomaterials in the introduction part. Authors are suggested to enhance the discussion on adsorption kinetics and consider the following papers for the same:

https://doi.org/10.1016/j.matlet.2022.131716

https://doi.org/10.1016/j.surfin.2022.102182

https://doi.org/10.1007/s11356-022-20743-8

Response: First we appreciate the good efforts from the respected reviewer in evaluating the manuscript. The given references are useful and have been cited in the revised manuscript. 

2. Authors need to improve the highlights as they are very generic and don’t exactly include the main points of their original work.

Response: Thank you for this comment. The highlights have been updated. 

3. The novelty aspects of this research paper need to be further modified and compared with the already present research (if any) to further enhance the overall impact.

Response: Thank you for this comment. Based to our best knowledge, this is the first report introducing TiO2-immoblized sand granules. However, table 02 introduces a comparison with other TiO2-based photocatalysts. 

4. How the reusability of catalyst can be assured? Please refer for some of the papers for a better explanation.

https://doi.org/10.1016/j.ijbiomac.2022.11.241

https://doi.org/10.1016/j.matlet.2022.131716

https://doi.org/10.1007/s11356-022-20743-8.

DOI: 10.1016/j.jmst.2021.01.051

Response: Thank you for this comment. The text has been updated with the support of the given references.

5. The paper in its current state has so many typos, technical and formatting related errors. There are certain space, grammar, and English related errors in the manuscript which are significantly ignored. Authors are suggested to proofread the manuscript thoroughly and eliminate the errors such as subscripts, superscripts, uniformity in presentation (Fig.Xa&b/Fig.X a and b).

Response: Thank you for this comment. The whole manuscript was carefully revised. 

5. Have authors tested it under light just to see effect of photocatalysis

Response: Thank you for this comment. For hydrogen generation, the experiments have been conducted under solar radiation as a source of visible light. However, in case of dye photo degradation, the experiments have been carried out under 2000 W Halogen lamp.

This explanation has been added in the revised manuscript. 

Reviewer #3: 

Please check typographical all the whole manuscripts. The author should rewrite the manuscript more concisely and succinctly, focusing on the obtained research results instead of rambling the textbook knowledge.

We strongly appreciate the great effort of the respected reviewer in evaluating our manuscript as well as his valuable comments. 

1. Introduction:

It should be shorter and more concise

Response: Thank you for this comment. The introduction section has been modified. 

2. Experimental part

It is not recommended to include basic knowledge from the textbook such as COD and BOD analysis procedures in the scientific article

Response: Thank you for this comment. The experimental section has been modified. 

3. The sections Results and discussion

Please focus on results and discussions in this section, do not include basic knowledge. 

There is no need to write too much about the concept, such as the concept of supercritical water.

Response: Thank you for this comment. The experimental section has been modified

It is not necessary to state the advantages and disadvantages and the purposes of the characterization of the produced catalyst analysis (EDX, XRD).

Response: Thank you for this comment. The text has been updated.

Fig 1 should be in section 2 Experimental part.

Response: Thank you for this comment. This figure introduces a conceptual illustration about the mechanism of formation of the proposed composite, so it is believed that this figure has to be in the results and discussion.

Table 1 should be used footnote for “Where COD is chemical oxygen demand, BOD is the biological oxygen demand, TSS is the total soluble suspended solids, TDS is the total dissolved solids, VSS is volatile suspended solids, total P is the total phosphorus, total N is the total nitrogen and Alk is the total alkalinity.” Paragraph.

Response: Thank you for this comment. The table has been modified

---

## [Decision Letter · Decision Letter 1]

6 Jun 2023

TiO2 NPs-immobilized Silica Granules: New Insight for Nano Catalyst Fixation for Hydrogen Generation and Sustained Wastewater Treatment

PONE-D-23-10591R1

Dear Dr. Barakat,

We’re pleased to inform you that your manuscript has been judged scientifically suitable for publication and will be formally accepted for publication once it meets all outstanding technical requirements.

Kind regards,

Van-Huy Nguyen, Ph.D.

Academic Editor

PLOS ONE

Additional Editor Comments (optional):

Reviewers' comments:

Reviewer's Responses to Questions

**Comments to the Author**

1. If the authors have adequately addressed your comments raised in a previous round of review and you feel that this manuscript is now acceptable for publication, you may indicate that here to bypass the “Comments to the Author” section, enter your conflict of interest statement in the “Confidential to Editor” section, and submit your "Accept" recommendation.

Reviewer #1: All comments have been addressed

Reviewer #2: All comments have been addressed

Reviewer #3: All comments have been addressed

2. Is the manuscript technically sound, and do the data support the conclusions?

Reviewer #1: Yes

Reviewer #2: Yes

Reviewer #3: Yes

3. Has the statistical analysis been performed appropriately and rigorously? 

Reviewer #1: Yes

Reviewer #2: Yes

Reviewer #3: Yes

4. Have the authors made all data underlying the findings in their manuscript fully available?

Reviewer #1: Yes

Reviewer #2: Yes

Reviewer #3: Yes

5. Is the manuscript presented in an intelligible fashion and written in standard English?

Reviewer #1: Yes

Reviewer #2: Yes

Reviewer #3: Yes

6. Review Comments to the Author

Reviewer #1: Manuscript ID: PONE-D-23-10591_R1

Title: TiO2 NPs-immobilized Silica Granules: New Insight for Nano Catalyst Fixation for

Hydrogen Generation and Sustained Wastewater Treatment

The authors addressed all reviewer comments, providing a comprehensive point-by-point response to every concern raised. Consequently, the manuscript is now eligible for acceptance in PLOS ONE.

Reviewer #2: (No Response)

Reviewer #3: The revision has been improved well. However, recheck some typos of English in the whole manuscript before publishing.

7. PLOS authors have the option to publish the peer review history of their article (what does this mean?). If published, this will include your full peer review and any attached files.

Reviewer #1: No

Reviewer #2: No

Reviewer #3: **Yes: **Lan Anh Phan Thi

---

## [Editor Report · Acceptance letter]

12 Jun 2023

PONE-D-23-10591R1 

TiO_2_ NPs-immobilized Silica Granules: New Insight for Nano Catalyst Fixation for Hydrogen Generation and Sustained Wastewater Treatment 

Dear Dr. Barakat:

I'm pleased to inform you that your manuscript has been deemed suitable for publication in PLOS ONE. Congratulations! Your manuscript is now with our production department. 

Kind regards, 

on behalf of

Dr. Van-Huy Nguyen 

Academic Editor

PLOS ONE